# Selected Factors of Experiencing Pregnancy and Birth in Association with Postpartum Depression

**DOI:** 10.3390/ijerph20032624

**Published:** 2023-02-01

**Authors:** Martina Bašková, Eva Urbanová, Barbora Ďuríčeková, Zuzana Škodová, Ľubica Bánovčinová

**Affiliations:** Department of Midwifery, Jessenius Faculty of Medicine in Martin, Comenius University in Bratislava, 03601 Martin, Slovakia

**Keywords:** antenatal classes, postpartum depression, attitudes toward pregnancy, support person during birth

## Abstract

Background: The aim of the study is to analyse the risk of postpartum depression using dimensions of perceived support (information, emotional, and physical), antenatal education (satisfaction and attitude), and attitude toward pregnancy (wanted or unwanted). Methods: A cross-sectional study was carried out among 584 postpartum women in two university birth centres in Slovakia. The Edinburgh Postnatal Depression Scale (EPDS) score was used. Descriptive statistics and analysis of variance, as well as logistic regression, were employed in the study. Found associations were adjusted for education level, type of birth, psychiatric history, and age. Results: As many as 95.1% of women reported their pregnancy as being wanted. Antenatal education, particularly satisfaction with it, showed a negative association with the EPDS score level. No significant differences in depression levels were found considering attitude toward pregnancy and perceived support. Conclusions: The study pointed out the significance of antenatal education to lower the risk of the postpartum depression. One of the important criteria of effective education is a woman’s subjective satisfaction with it.

## 1. Introduction

Depression is one of leading causes of morbidity among women, and its prevalence is two-fold greater than that in men. Maternal postpartum depression is considered to be one of the most common and disabling complications of childbirth, but very often it is undiagnosed and untreated [1,2].

Postpartum depression (PPD) and nonperinatal major depression share the same diagnostic criteria: a combination of depressed mood, loss of interest, anhedonia, fatigue, feelings of guilt or worthlessness, sleep and appetite disturbance, psychomotor disturbance, and suicidal thoughts, which are present during the same two-week period and are a change from previous functioning [2]. According to the American Psychiatric Association 2013, a major depressive episode occurring within pregnancy or 4 weeks after birth is depression with a perinatal onset. Diagnostic standards vary, but many define PPD as a disorder occurring anytime within the first year, and mostly within 6 months, after giving birth [3]. Evidence on the predictors and outcomes of PPD mainly use self-reporting measures of elevated depressive symptoms rather than diagnostic interviews [4]. Active screening and following treatment based on cooperation between gynaecology−obstetrics and psychiatry is the major method of postpartum depression prevention [5].

In developed countries, the prevalence of postpartum depression ranges from 5.5% to 34.4% between birth up to four weeks, 2.6–35.0% for four to eight weeks, 2.9–25.5% for six months, and 6.0–29.0% for 12 months postpartum. The prevalence also significantly varies across countries. The prevalence of postpartum depression in developing countries is higher than in developed ones, ranging from 12.9% to 50.7% between birth up to four weeks, 4.9–50.8% for four to eight weeks, 8.2–38.2% for six months, and 21.0–33.2% for one year postpartum [6]. 

Many factors play a role in the development of postpartum depression; besides biological factors, psychosocial factors also play an important role, such as a history of psychiatric illness prior to pregnancy, a low level of social support, and domestic violence during pregnancy or after delivery [5]. In addition, unwanted or unplanned pregnancy may be a factor for developing a negative attitude toward pregnancy in general. These women have a low level of social support and another common consequence of unwanted/unplanned pregnancy is a low adjustment to the parental role and a higher risk for the development of postpartum depression [7]. On the other hand, there are studies that show that women whose first onset of depression was in the postpartum period were at increased risk for future development of postpartum depression, but not for depressive episodes that occurred outside of the postpartum period. In comparison, women whose PPD occurred as a recurrent depression were at increased risk for depressive episodes occurring outside of the postpartum period, but not in the postpartum period [8].

The birth experience is strongly affected by women´s personal expectations, quality of caregiving support during childbirth, and the birth environment [9,10].

The evidence has demonstrated that women who attended childbirth preparation classes increased their confidence about their ability to deal with childbirth, and their scores of fear of childbirth, anxiety, and depression were significantly lower than those who had not [11,12]. Participation in childbirth preparation classes should be recommended to all women who are going to give birth for the first time, as this type of preparation was found to significantly decrease the level of fear for childbirth [13]. Another factor that can affect the onset of PPD is having a supportive person around during childbirth. This support came most often from companions of the woman’s choice, such as her partner, mother, or closest friends. Companionship was found to help women to have a positive birth experience in general, and the World Health Organization [14,15] discusses the significance of continuous support during childbirth. Postpartum depression could be lower in women supported in labour, but we cannot be sure of this statement because these studies are difficult to compare [16,17]. Women mostly remember the quality of healthcare during the childbirth, and it can also suppress effect of many other factors such as pain or difficulty during birth, and result in a positive birth experience [3]. Association between the risk of PPD and mode of delivery, as well as other birth events (medical interventions, complications, and delayed mother−infant contact) has been explored; however, the conclusions are incoherent [18,19]. Independently of the course of labour, healthcare providers can help ensure supportive care and encourage women´s confidence, respect, privacy, and feeling of safety [3]. It is well known that pregnancy and delivery is positively influenced by the support, antenatal education, women´s attitude toward it [16,20,21]. The question is whether these above-mentioned issues can contribute to reducing the risk of postpartum depression. The main aim of the study was to explore whether antenatal education (satisfaction with it and attitude toward it), attitude toward pregnancy (wanted or unwanted), and support provided during birth (information, emotional, and physical) are factors significantly associated with the levels of postpartum depression symptoms. The study follows on from our previous paper dealing with the basic possible factors influencing the risk of the postpartum depression [22]. Our findings can contribute to understanding the role of midwifery interventions during pregnancy in the development of postpartum depression and to propose effective ways to reduce its negative impact.

## 2. Methods

### 2.1. Data Collection and Sample

A cross-sectional research design was used in this study. We used a convenient sample design including postpartum women in two university birth centres in Slovakia: the first one situated in Martin, Central Slovakia, and the second one in Bratislava, the capital. Data were collected on the 2nd to the 4th postpartum day during the hospital stay of the mothers. The inclusion criteria were 2nd to the 4th day postpartum and informed consent, and the exclusion criteria were stillbirth and a history of severe psychiatric disorder (psychotic disorder). Participants filled out a paper−pencil questionnaire administered by qualified midwifes. Data collection was carried out from September 2019 to April 2020. 

We approached 710 postpartum women (Table 1). From them, 584 agreed to participate in the study (82.3% response rate). Their ages ranged from 16 to 44 years (mean 30.6 ± 4.9 years). Among the participants, there were 431 (73.8%) vaginal births and 153 (26.2%) operative births. The sample included 346 (59.9%) primiparas, 184 (31.8%) secundiparas, and 48 (8.3%) multiparas. Considering education, 13 (2.2%) mothers achieved basic education, 231 (39.6%) middle education (high school with or without graduation), and 339 (58.2%) high education (university degree). Of the 584 participants, 381 (65.2%) were recruited in Martin and 203 (34.8%) in Bratislava.

### 2.2. Measuring Instruments

The Edinburgh Postnatal Depression Scale (EPDS) was used to assess the symptoms of postpartum depression. It contains 10 items enquiring about common depressive symptoms [23]. Each question has four choices per statement (rated 0−1−2−3), resulting in a total score ranging from 0 to 30. A higher score indicates a higher level of depression symptoms. According to the EPDS manual, second edition (2014), different cut-off points of the EPDS were used. In our study, we used a cut-off point of 10 points or more for indicating elevated levels of depressive symptoms. Internal consistency (Cronbach alpha) in the present research was 0.84. Permission to use EPDS was obtained from the Royal College of Psychiatists (UK). We performed a back-translation of the Slovak version of the questionnaire with additional assessment of the accuracy of the translation. Besides the EPDS items, in the first phase, the questionnaire used also included sociodemographic and medical history data such as age and education, type of birth, parity, preterm birth, history of perinatal loss, and complications during pregnancy. Psychometric properties of the Slovak version of EPDS were found to be satisfactory [23].

Subjective perception of the support received during birth (from a partner or another person) was evaluated using three questions. Each question was focused on one dimension of support: informational, emotional, and physical (e.g., “My birthing partner provide me with the emotional support during birth”). Informational support questions included issues such as help with communication with staff, help with obtaining information from staff, and interpreting women’s requests to staff. Emotional support questions included encouragement, help with relaxation, providing company, expression of support by hugging, and providing feelings of safety. Physical support questions focused on providing help with changing positions, showering, massage, and assistance with daily necessities. Only women who had a support person during their birth were asked to fill out questions focused on subjective perception of the support received during birth, thus 409 women were eligible for filing out these questions.

Subjective perception of the information provided by antenatal education was evaluated using two questions. The first question was focused on subjective satisfaction with the level of information about birth-related issues (“I feel I have enough information about birth and I was prepared for the birth”). The second question asked about attitude toward antenatal education (“I felt it was better not to have birth-related information before birth”); a positive answer to this question indicated a negative attitude toward antenatal education, and a negative answer indicated a positive attitude toward antenatal education. Each question was rated on five point Likert scale (1, strongly disagree; 2, disagree; 3, neither agree nor disagree; 4, agree; 5, strongly agree). However, for the purposes of the statistical analyses, categories were merged into three (positive evaluation, neutral, and negative evaluation), as the number of answers in categories 1 and 2 were very low.

Attitude toward pregnancy was assessed using a question formulated as follows: “How would you describe your attitude toward your pregnancy?”. Participants were able to choose one from following answer possibilities: 1, planned, wanted; 2, unplanned, wanted; 3, unplanted, unwanted; 4, planned, unwanted.

Questions on demographic and anamnestic information (age, education, parity, etc.) were included in the research questionnaire as well.

### 2.3. Statistical Procedures

Descriptive statistics and analysis of variance, as well as logistic regression, were employed in the study. Logistic regression models were used in order to examine the effect of attitudes toward antenatal education, attitudes toward pregnancy, and perceived stress in pregnancy on the occurrence of a high level of postpartum depression symptoms. In the first step, the crude effects of independent variables (stress level, attitude toward pregnancy, attitude toward antenatal education, and satisfaction with the antenatal education) on postpartum depression symptoms were computed using the lowest levels as a reference category. Next, the model was adjusted for education (high, middle, and low), type of birth (spontaneous vs. operative), psychiatric history (positive vs. negative), and age. Analyses were performed using SPPS 26.0 for Windows.

### 2.4. Ethical Issues

All of the participants were thoroughly informed about the project aims and ethical issues (anonymity, personal data protection, and voluntary participation). A consent form was signed by each participant prior to their participation. Ethical approval was obtained from the Ethical committee of the Jessenius Faculty of Medicine in Martin, Slovakia, no. EK 36/2018.

## 3. Results

More than half of our respondents claimed that their pregnancy was planned/wanted (Table 2). Attitude toward antenatal education was mostly positive, but 19.4% of respondents expressed a negative attitude. In addition, satisfaction with antenatal education was high in most cases (55.8%), and a low satisfaction was expressed by 14.7% respondents. In general, support during childbirth (emotional, physical, and informational) was perceived in a positive sense.

Both the attitude towards the provided antenatal education and the level of satisfaction with it were significantly associated with the EPDS score (Table 3). Thus, women reporting positive attitudes and satisfaction had the lowest EPDS score levels. Considering other analysed factors, no significant differences were found.

No significant differences in likelihood of experiencing elevated depressive symptoms postpartum were found according to participant’s positive or negative attitude toward pregnancy (wanted vs. unwanted pregnancies). Women with negative attitude toward antenatal education had higher likelihood of feeling depressed during postpartum compared to the women with positive attitude toward antenatal education (Table 4). The association remained significant even after controlling for the effect of the history of a psychiatric disease, type of birth, age and education (OR = 2.27). On the other hand, no significant differences in likelihood of experiencing elevated postpartum depressive symptoms were found based on satisfaction with antenatal education.

Value of the Hosmer and Lemeshow Test was 4.26 (*p* = 0.833) for the first model, and 4.55 (*p* = 0.805) for second model adjusted for possible covariates, which indicates a good model fit in both models.

## 4. Discussion

According to our results, consistent with previous studies [22,24], postpartum depression affects a considerably high proportion of women. An overwhelming majority of women expressed positive views towards support during their childbirth. Similarly, the majority of them reported positive attitudes toward antenatal education. These results indicate that postpartum depression presents a significant issue deserving attention. Moreover, the results pointed towards the positive role of provided support and education. We positively consider that the overwhelming majority of the women reported their pregnancy as wanted, being an essential precondition of the physiological course of the postpartum period.

A numerous range of factors contribute to the development of postpartum depression. They undoubtedly include insufficient social support either from a partner or a family, as well as whether the pregnancy was planned or not [7].

Unplanned pregnancy probably influences the entire life and career plans of women, and thus can predispose to the development of depression during the postpartum period [25]. Some studies describe a two-fold higher probability of the development of the postpartum depression in women not planning their pregnancy compared with those planning it [26]. However, our study did not support this view and did not show significant links between the occurrence of signs of postpartum depression and the pregnancy intention (planned/wanted vs. unplanned/unwanted).

The subjective perception of support during birth was perceived very positively in our group in all areas of support, i.e., informational, emotional, and physical. The area of emotional support was best rated, but the differences were very small (Table 2). Emotional support is one of the most important benefits provided by a support person for all types of assistance during birth. In particular, a feeling of security is important for women [16], as it positively affects the experience of pain, ensures that women remain focused on childbirth [15], and is assumed to have a significant role in maintaining mental well-being after birth [27]. Most of the women from the group confirmed the presence of a support person who provided positive support during childbirth, but a significant association between birth support and PPD was not confirmed. However, it is possible to assume that social support during pregnancy, childbirth, and in the postpartum period is generally a protective factor for the development of PPD [15,28,29]. Moreover, support during childbirth is also involved in overall birth satisfaction, where the link with the risk of PPD has been confirmed [22].

Our study showed an increased risk of postpartum depression symptoms in women not satisfied with their antenatal education compared with the women who were satisfied. This indicates that a positive view of and satisfaction with antenatal education may reduce postpartum depressive symptoms. However, an earlier study published in 2001 [30] carried out among primiparas did not show such an influence, and rather indicated positive effect of education on the overall health awareness of women. In accordance with our results, Shimpuku et al. [31] described a positive effect of high-quality antenatal education for preventing postpartum depression and increasing parental self-confidence.

Antenatal education can provide women and their partners with numerous pieces of useful information about changes in mental health in the postpartum period and should be focused on a range of associated emotional problems. It is not easy to disclose the first symptoms of postpartum depression and women often do not talk about their feelings of unhappiness after their delivery. Through aimed educational sessions, women and their partners can become better aware of postpartum depression symptoms and look for timely assistance [32]. Antenatal education is probably more effective than postpartum education. Postpartum women are more tired and too much time spent on the education can actually result in negative effects [33]. Therefore, antenatal education or education during the hospital stay and shortly after it can be more effective. Pregnancy in psychological terminology is referred to as a developmental transition, which is characterized by the accumulation of changes in the psychological field. The way of dealing with these changes depends on the woman’s personality, identification with the female role, relationship with the unborn child, attitudes towards her own pregnancy, specific socio-economic conditions, and the physical condition of the future mother. In general, pregnant women are more sensitive; increasingly anxious; and are prone to depression, moodiness, and aggression. Attending childbirth preparation classes is the best choice and a prerequisite for a positive, physiological course of childbirth. The aim of the childbirth preparation classes is to reduce fear, concerns, and anxiety through an appropriate and individual approach to the mother, as well as to create a positive attitude towards pregnancy and childbirth [3].

Prenatal care and childbirth preparation classes should be an essential part of the current health care for women, namely mothers in the European Union. In general, the prenatal care program deals with diseases screening, supporting a healthy lifestyle, and assessing the state of the future child. This care is aimed at reducing perinatal morbidity and mortality of the mother and child [12]. The importance of childbirth preparation classes for pregnant women is in obtaining information, especially instructions on how to cope with various situations during childbirth. It is also perceived as a prevention, as proper instruction and childbirth preparation classes can often prevent various complications during pregnancy. Preparation courses for childbirth have several positives outcomes: they reduce the feeling of anxiety and the perception of labour pain, reduce the possibility of using pharmacotherapy, shorten the duration of childbirth, increase psychological benefits of experiencing childbirth and satisfaction with cooperation with partner, strengthen family relationships, help develop a positive relationship with the new born, and create a prerequisite for a healthy course for the postpartum period.

High quality childbirth preparation classes have a positive effect on improving family relationships, family planning, health habits, and stress management; reducing anxiety; and, last but not least, encourage postpartum adaptation and successful breastfeeding.

Childbirth preparation classes (CPCs) should meet the basic neurophysiological and obstetric requirements and conditions. They consist of preparing pregnant women to cope mentally and physically with the body during pregnancy and teach them to cooperate during childbirth. The importance of childbirth preparation classes must also be assessed from the point of view of the people accompanying the mothers—partners/spouses. The quality of CPC will teach partners the basics of each stage of labour and techniques to help mothers during each stage of childbirth. The presence of an informed accompanying person in the delivery room, who has realistic expectations, thus becomes a help for the mother and not a burden for the medical staff. Our findings confirmed this aspect, as the overwhelming majority of respondents considered information, emotional and physical support during birth, and antenatal education positively (Table 2).

Childbirth preparation classes (CPCs) and proper awareness are effective methods for reducing discomfort and can also reduce the risk of complications. Previous studies have shown a decreased incidence of sudden indications for caesarean section among women undergoing childbirth preparation classes. On the other hand, CPC does not affect the demand for epidural anaesthesia from mothers [34].

The methodological limitations of the study include the fact that an analysis of outliers and missing data was not carried out. Data on differences in the attendance of antenatal classes in our study were not collected, which is a factor that could influence the results of our analysis. The possibility of generalization of our results to the whole population of Slovak birthing women might be influenced by the fact that the study sample in our research more often included highly educated women and primparas compared with the entire population of birthing women in Slovakia. Taking into consideration the importance of satisfaction with provided education in the study, further research should be focused on the factors determining this satisfaction.

## 5. Conclusions

The study analysed the links between factors related to pregnancy, such as support during delivery, satisfaction with education, attitude towards pregnancy, and the development of postpartum depression. Attitude towards pregnancy, i.e., if it was planned or unplanned, did not show significant effects on depression. On the other hand, women reporting dissatisfaction with the education provided showed a higher EPDS score, indicating an increased risk of postpartum depression. However, although the majority of women expressed satisfaction with the education, there was a significant proportion reporting only neutral or low-level satisfaction. Thus, midwives should pay better attention to the content and form of the education so as to achieve the highest possible satisfaction among pregnant women. Moreover, well composed education can help women to better recognize the first signs of the development of depression and to report them to healthcare staff. Therefore, midwives can implement appropriate interventions in time as one of the ways of reducing the development of depression, potentially leading to further negative consequences.

## Figures and Tables

**Table 1 ijerph-20-02624-t001:** Demographic and perinatal description of the study participants.

Variable		Frequency	Percentage
Education	Basic	13	2.2
	Middle	231	39.6
	High	339	58.1
	Missing data	1	0.2
Family status	Single−no partner	5	0.9
	Single−with partner	201	34.4
	Married	366	62.7
	Widowed	12	2.1
Financial situation	Very good	251	43.0
	Quite good	307	52.6
	Not very good	23	3.9
	Missing data	3	0.5
History of psychiatric disease	No	556	95.2
	Yes	27	4.6
	Missing data	1	0.2
Support person during birth	Partner	376	64.4
	Friend/relative	31	5.3
	Dula	2	0.3
	None	175	30.0
Parity	Primipara	346	59.3
	Multipara	232	39.7
	Missing data	6	1.0
Type of birth	Spontaneous	431	73.8
	Instrumental	16	2.7
	Planned SC	51	8.7
	Acute SC	86	14.7
Total sample		584	100

**Table 2 ijerph-20-02624-t002:** Subjective perception of a support during birth and antenatal education, attitudes toward pregnancy, and levels of postpartum depression among the study participants.

Subjective Perception of Factors Related to a Birth	Frequency	Percentage
Information support		
Negative	21	5.1
Neutral	50	12.2
Positive	331	80.9
Missing	7	1.2
Total	409	100.0
Emotional support		
Negative	0	-
Neutral	4	1.0
Positive	400	97.80
Missing	5	1.22
Total	409	100.0
Physical support		
Negative	13	3.18
Neutral	13	3.18
Positive	376	92.6
Missing	7	1.17
Total	409	100.0
Satisfaction with antenatal education		
Low	86	14.72
Neutral	166	28.42
High	326	55.82
Missing	6	1.03
Total	584	100.0
Attitude toward antenatal education		
Negative attitude	113	19.3
Neutral	119	20.4
Positive attitude	347	59.4
Missing	5	0.9
Total	584	100.0
Attitude toward pregnancy		
Planned, wanted	398	68.2
Unplanned, wanted	157	26.9
Unplanned, unwanted	5	0.9
Planned, unwanted	24	4.1
Total	584	100.0
EPDS score ≤ 10		
Yes	96	16.4
No	481	82.4
Missing	7	1.2
Total	584	100.0

**Table 3 ijerph-20-02624-t003:** Analysis of variance for differences in total EPDs scores between different groups of participants according to antenatal education, attitudes toward pregnancy, and support during birth.

		EPDS Score	ANOVA/*t*-Test	Sheffe Post Hoc Test: Significant Differences between Groups
Attitude toward pregnancy	Unwanted pregnancy	5.64 (±4.90)	−1.40	-
	Wanted pregnancy	5.08 (±4.12)		
Attitude toward antenatal education	Positive	4.74 (±4.10)	**5.69 ****	1–3
	Neutral	5.76 (±4.50)		
	Negative	6.17 (±4.76)		
Satisfaction with antenatal education	High	4.90 (±4.40)	**4.64 ****	1–3
	Neutral	5.21 (±4.15)		
	Low	6.49 (±4.33)		
Informational support during birth	Positive	5.08 (±4.23)	0.78	-
	Neutral	5.86 (±3.46)		
	Negative	6.15 (±5.57)		
	No support person	5.23 (±4.58)		
Emotional support during birth	Positive	5.26 (±4.26)	0.53	-
	Neutral	3.00 (±2.16)		
	Negative	-		
	No support person	5.24 (±4.60)		
Physical support during birth	Positive	5.16 (±4.24)	1.63	-
	Neutral	6.77 (±4.73)		
	Negative	5.76 (±4.40)		
	No support person	5.23 (±4.57)		

Statistically significant effect is in bold (*p* ≤ 0.001 **).

**Table 4 ijerph-20-02624-t004:** Logistic regression analysis- risk of having higher levels of postpartum depression symptoms in groups with regard to antenatal education and attitudes toward pregnancy.

Elevated Depression Symptoms		Crude Effect OR (95% CI)	Adjusted Effect * OR (95% CI)
Attitude toward pregnancy	Unwanted pregnancy	1	1
	Wanted pregnancy	0.79 (0.46–1.35)	0.86 (0.49–1.52)
Attitude toward antenatal education	Positive	1	1
	Neutral	1.49 (0.80–2.79)	1.60 (0.84–3.06)
	Negative	**2.09** (1.16–3.78)	**2.27** (1.23–4.18)
Satisfaction with the antenatal education	High	1	1
	Neutral	0.86 (0.47–1.59)	0.78 (0.42–1.45)
	Low	1.15 (0.59–2.21)	1.10 (0.56–2.18)

* Effect adjusted for psychiatric anamnesis, type of birth, age and education. Statistically significant effect is in bold (*p* < 0.05).

## Data Availability

The data presented in this study are available upon request from the corresponding author. The data are not publicly available due to ethical and privacy restrictions.

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
