# Peer review of "Selected Factors of Experiencing Pregnancy and Birth in Association with Postpartum Depression"

_ijerph, 2023, doi:10.3390/ijerph20032624_

Round 1

Reviewer 1 Report

      The main objective of this study is to analyse risk of postpartum depression by dimensions of provided support (information, emotional and physical), antenatal education (satisfaction and attitude) as well as attitude toward pregnancy (wanted or unwated)”. The topic is interesting, however, need some improvement.

1.     In abstract it’s better inform about methods of data analysis.

2.     Line 9, “wanted or unwated”. Please correct it.

3.     Line 16 in abstract, the authors mentioned that “No significant differences were found considering other analysed factors”. It’s not clear. Please elaborate more.

4.     In abstract the authors mentioned “Conclusion: Well provided antenatal education can contribute to lower risk of the postpartum depression. Midwives should pay better attention to its content and form to achieve as much as possible satisfaction among pregnant women”. This conclusion is not strong enough.

5.     How did the authors calculate 584 as sample size?

6.     What is the main question of the research?

7.     Research hypotheses are not clear.

8.     Line 97 “Data collection was performed from September to April 2020”. Please rewrite the sentence.

9.     Why authors didn’t use Structural Equation Modeling (SEM), or PLS?

10.  What is the novelty of the research?

11.  There is lack of information about reliability and validity of the questionnaire.

12.  There is no analysis of outliers and missing data.

13.  There are no theories behind research variables’ measurement including “Attitude toward pregnancy”, “subjective perception of the support”, and “subjective perception of the information”

14.  There is lake of information about “limitation of the study” and “suggestion for future studies”

Author Response

# Reviewer 1:

The main objective of this study is to analyse risk of postpartum depression by dimensions of provided support (information, emotional and physical), antenatal education (satisfaction and attitude) as well as attitude toward pregnancy (wanted or unwated)”. The topic is interesting, however, need some improvement.

  1. In abstract it’s better inform about methods of data analysis.

Author’s response: Information was added to the abstract.

  1. Line 9, “wanted or unwated”. Please correct it.

Author’s response: We fixed this mistyping.

  1. Line 16 in abstract, the authors mentioned that “No significant differences were found considering other analysed factors”. It’s not clear. Please elaborate more.

Author’s response: Information was added to the abstract.

  1. In abstract the authors mentioned “Conclusion: Well provided antenatal education can contribute to lower risk of the postpartum depression. Midwives should pay better attention to its content and form to achieve as much as possible satisfaction among pregnant women”. This conclusion is not strong enough.

Author’s response: We rephrased the conclusions in the abstract as: “The study pointed out significance of antenatal education to lower risk of the postpartum depression. One of the important criteria of effective education is a satisfaction with it among pregnant women.

  1. How did the authors calculate 584 as sample size?

Author’s response: Total number of 584 participants were recruited in the research sample based on convenient sampling method. We added this information into the text (line 112 and 113)

  1. What is the main question of the research?

Author’s response: This information was added to the introduction (line 92 and 93 in corrected manuscript).

  1. Research hypotheses are not clear.

Author’s response: The information was added to the introduction (line 92 and 93 in corrected manuscript).

  1. Line 97 “Data collection was performed from September to April 2020”. Please rewrite the sentence.

Author’s response: This sentence was rewritten (line 110 in corrected manuscript)

  1. Why authors didn’t use Structural Equation Modeling (SEM), or PLS?

Author’s response: Given the main research question of the study (association between antenatal education, attitude toward pregnancy, and support provided during birth and levels of postpartum depression), ANOVA nad logistic regression was chosen as statistical methods.

  1. What is the novelty of the research?

Author’s response: The study showed decreased risk of the postpartum depression symptoms in women satisfied with an antenatal education. It indicates that a positive view and satisfaction with antenatal education can be considered as an important criterion of its effect to reduce postpartum depressive symptoms.

  1. There is lack of information about reliability and validity of the questionnaire.

Author’s response: Information on reliability and validity of the Slovak version of the EPDS were added in methods section of the manuscript (Line 129)

  1. There is no analysis of outliers and missing data.

Author’s response: We have added this fact to the list of methodological limitations in the discussion section of the revised manuscript (line 313)

  1. There are no theories behind research variables’ measurement including “Attitude toward pregnancy”, “subjective perception of the support”, and “subjective perception of the information”

Author’s response: We added a rationale for the study supported by respective references “It is well known that a course of pregnancy and delivery is positively influenced by provided support, antenatal education as well as attitude toward it [Hodnett et al., 2013, Bohren et al., 2017]. The question is, if these above-mentioned issues can contribute to reduce risk of postpartum depression. (line 85)

  1. There is lake of information about “limitation of the study” and “suggestion for future studies”

Author’s response: We have added paragraph on methodological limitations and suggestion for future research in the discussion part of revised manuscript (line 313)

Reviewer 2 Report

Thank you for the opportunity to review this paper which has the potential to inform practice by determining factors which affect postnatal depression. This is the second paper using the findings from 584 women during the postnatal period. This paper aimed to understand if support, antenatal education and attitude to pregnancy influence postnatal depression. Results showed that satisfaction with antenatal education was negatively associated with postnatal depression scores. This paper shows associations between perceived satisfaction of antenatal education and lowers the risk of postpartum depression.

Overall the paper is well structured and written with appropriate methodology but requires significant changes to improve the use of the English language. This will make the paper easier to read and understand and may be the reason for some of the confusion and comments I have made.

I have some comments which I think will enhance the paper.

Abstract

Appropriate content.

Introduction

1)      Line 40 and 44, ‘…in less than four weeks’ do you mean at four weeks or between birth and four weeks? Please clarify.

2)      Line 42, ‘..in 12 months’ do you mean at 12 months’? Please clarify.

3)      Line 45, ‘… in the first year’, do you mean ‘at’ one year postpartum or the overall rate for PPD during the first year after birth. Please clarify.

4)      Line 59, ‘There has been less attention period..’? do you mean less attention paid’. Please clarify. Also you state that there has been less attention in associations between childbirth experience and the risk of development of PPD. I would disagree with this statement. There have been many papers reporting associations between birth experiences and PPD. In fact your previous publication states that lower satisfaction with childbirth may increase the risk of developing PPD’

5)      Line 87. I am not sure midwifery interventions during pregnancy will ever prevent PPD. They may reduce it. Please think about rephrasing this statement.

Methods

6)      Line 91-94. How did the midwives decide who could be recruited. What were the inclusion and exclusion criteria for recruitment?

7)      Line 95-97. The methods describe recruitment on 2nd to 4th day by ‘trained midwives’ and the completion of questionnaires by participants. Were these paper copies? The you go on to state that participants were contacted to fill in an online questionnaire. It is not clear if these are two separate questionnaires. Please could you clarify if they are the same questionnaire or two different ones or did participants have a choice. If it is a follow up questionnaire when and where were these completed.

8)      Line 95. Does ‘trained midwives’ mean they were trained to administer the study questionnaire and trained in the study protocol or are they qualified midwives as opposed to student midwives? Please could you clarify.

9)      Line 103. I know every education system is different but could you explain the educational standards, what is basic, middle and higher educations? Is higher education graduate level and above or master/PhD. This would make it easier to compare against the findings of other publications.

10)   Line 104. It is unusual to start a sentence with a number, you could say ‘Of the 584 participants, 381 ….’ 

11)   Line107. This section describes the screening tools. Demographic data must also have been collected as it is presented in the results section. Could you add a sentence or two to explain how the other data was collected. Was this part of the questionnaire or was it collected by the midwives from the maternity records?

12)   Line 109. I am not sure why there is a reference in the manuscript which is different from the rest of the referencing style (Cox et al. 2014) is in the text but not in the reference list.

13)   Line 129. What is meant by ‘handling things’?

14)   Line 161. What is an ‘informed consent letter’? They are usually called consent forms.

Tables

15)   In Table 1, ‘anamnesis’ is a very unusual word to use and I am not sure what this is relating to. The word is not used anywhere else in the manuscript. Could you explain the word somewhere in the text or use a different word?

16)   Table 1 and 2, in the title/legend, what do you mean by ‘in a research sample’? do you mean the  study participants?

17)   Table 3, there is a comma which I think should be a full stop for informational support.

Results

18)   If the main aim of the study is looking at antenatal education, did you collect the number of participants who took part in antenatal education, or was this in the inclusions criteria that they must have attended education in the antenatal period? Also I notice you included primiparous and multiparous women in the study. Did more primiparous mothers attend classes than multiparous women? Did you look at the comparison between these groups. I would think there would be a difference in expectations and attendance between the two groups and this may alter the findings.

Discussion

19)   The title suggests there may be several factors during pregnancy and birth that are associated with PPD. The discussion seems to focus mostly on the need for antenatal education, which were statistically significant findings. The manuscript does not report on the informational and emotional support. I know these were not significant but they could still be compared to previous research. Often emotional support during birth is seen as positive and a lack of support a factor in postnatal depression. Were there differences in your study which may account for this conflict?

20)   Line 226. What is meant by ‘can provide pair….’ Please could you clarify.

21)   Line 233. Is there any research to back up the statement ‘too much time spent on postnatal education can result in negative effects’? Is this related to the findings of reference 27? If not is a bold statement to make without evidence.

22)   Line 235. I am unsure what you mean by ‘Pregnancy is psychological terminology in referred as a developmental crisis…’. Please could you clarify what you are saying.

23)   Line 244. I think the work ‘eliminate’ is far too strong a word to use. To eliminate means remove completely. Antenatal education will not be able to eliminate PPD. Reduce would be a more appropriate word.

Conclusion

24)   As in the discussion, the conclusions does not refer to information and emotional support during labour.

25)   Line 291. ‘Therefore, midwives can soon….’. I feel this is quite a bold statement to make, can worsening depression be prevented by well-structured antenatal classes? You could mention that antenatal classes may be one way of reducing the symptoms or supporting women’s mental health.

References

26)   There are a large number of references that are over 5 years (the time period suggested by the journal). I understand the need for the use of seminal papers and references for screening tools which are used which may be decades old. Please could you review and update references where possible.

There are many references which could be updated, for example:

Adams, 2012. There many papers looking at mode of delivery and PPD. E.g. Baba eta l 2021     looked at mode of delivery and postpartum depression in Japan and a systematic review and meta analysis by Sun et al 2021 also reived this topic.

Hayes, 2001. There are many papers looking at antenatal education and perinatal depression. A recent article looked at antenatal education in primiparous women. Cankaya et al 2021.

Bergstrom, 2009. There are many papers with antenatal education on epidural rates. There has recently been a systematic review and meta-analysis. Hong et al 2021. Although this contradicts the paper you have referenced relating to epidural rates.

Author Response

# Reviewer 2:

Thank you for the opportunity to review this paper which has the potential to inform practice by determining factors which affect postnatal depression. This is the second paper using the findings from 584 women during the postnatal period. This paper aimed to understand if support, antenatal education and attitude to pregnancy influence postnatal depression. Results showed that satisfaction with antenatal education was negatively associated with postnatal depression scores. This paper shows associations between perceived satisfaction of antenatal education and lowers the risk of postpartum depression.

Overall the paper is well structured and written with appropriate methodology but requires significant changes to improve the use of the English language. This will make the paper easier to read and understand and may be the reason for some of the confusion and comments I have made.

I have some comments which I think will enhance the paper.

Abstract

Appropriate content.

Introduction

  • Line 40 and 44, ‘…in less than four weeks’ do you mean at four weeks or between birth and four weeks? Please clarify.

Author’s response: We rephrased the term as “between birth up to four weeks”.

  • Line 42, ‘..in 12 months’ do you mean at 12 months’? Please clarify.

Author’s response: Yes, we meant “at 12 months”. We fixed it.

  • Line 45, ‘… in the first year’, do you mean ‘at’ one year postpartum or the overall rate for PPD during the first year after birth. Please clarify.

Author’s response: We meant one year postpartum. We fixed it.

  • Line 59, ‘There has been less attention period..’? do you mean less attention paid’. Please clarify. Also you state that there has been less attention in associations between childbirth experience and the risk of development of PPD. I would disagree with this statement. There have been many papers reporting associations between birth experiences and PPD. In fact your previous publication states that lower satisfaction with childbirth may increase the risk of developing PPD’

Author’s response: We erased the respective sentence.

  • Line 87. I am not sure midwifery interventions during pregnancy will ever prevent PPD. They may reduce it. Please think about rephrasing this statement.

Author’s response: We aggree with this reasonable comment. We used the term “to reduce its negative impact”.

Methods

  • Line 91-94. How did the midwives decide who could be recruited. What were the inclusion and exclusion criteria for recruitment?

Author’s response: Inclusion and exclusion criteria were added to the revised manuscript (Line 106, methods section)

  • Line 95-97. The methods describe recruitment on 2nd to 4th day by ‘trained midwives’ and the completion of questionnaires by participants. Were these paper copies? The you go on to state that participants were contacted to fill in an online questionnaire. It is not clear if these are two separate questionnaires. Please could you clarify if they are the same questionnaire or two different ones or did participants have a choice. If it is a follow up questionnaire when and where were these completed.

Author’s response: Participants in this phase of the research were filling up a paper-pencil questionnaires. Results of the follow up study were not analysed in this paper. We have clarified this in the methodological section of the revised manuscript (line 108)

  • Line 95. Does ‘trained midwives’ mean they were trained to administer the study questionnaire and trained in the study protocol or are they qualified midwives as opposed to student midwives? Please could you clarify.

Author’s response: We meant qualified midwives, this was corrected also in the revised manuscript (line 108)

  • Line 103. I know every education system is different but could you explain the educational standards, what is basic, middle and higher educations? Is higher education graduate level and above or master/PhD. This would make it easier to compare against the findings of other publications.

Author’s response: Middle education was high school with or without graduation, high education was university degree (we have clarified this in the revised manuscript- line 116)

  • Line 104. It is unusual to start a sentence with a number, you could say ‘Of the 584 participants, 381 ….’

Author’s response: Thank you for this comment, we have corrected the mentioned sentence in the revised manuscript (line 118)

  • This section describes the screening tools. Demographic data must also have been collected as it is presented in the results section. Could you add a sentence or two to explain how the other data was collected. Was this part of the questionnaire or was it collected by the midwives from the maternity records?

Author’s response: Questions on demographic and anamnestic data were included in the research questionnaire. We have added this information in the revised manuscript (line 165)

  • Line 109. I am not sure why there is a reference in the manuscript which is different from the rest of the referencing style (Cox et al. 2014) is in the text but not in the reference list.

Author’s response: Thanks for this comment. We overlooked the reference, it has been fixed.

  • Line 129. What is meant by ‘handling things’?

Author’s response: We used instead the term “assistance with daily necessities”.

  • Line 161. What is an ‘informed consent letter’? They are usually called consent forms.

Author’s response: Thank you for this comment, we have corrected the mentioned sentence in the revised manuscript (line 180)

Tables

  • In Table 1, ‘anamnesis’ is a very unusual word to use and I am not sure what this is relating to. The word is not used anywhere else in the manuscript. Could you explain the word somewhere in the text or use a different word?

Author’s response: This term in Table 1 was replaced with the “history of psychiatric disease”

  • Table 1 and 2, in the title/legend, what do you mean by ‘in a research sample’? do you mean the study participants?

Author’s response: Thank you for this comment, we have corrected titles of the Table 1 and 2 in the revised manuscript

  • Table 3, there is a comma which I think should be a full stop for informational support.

Author’s response: This mistake was corrected in Table 3 in the revised manuscript.

Results

  • If the main aim of the study is looking at antenatal education, did you collect the number of participants who took part in antenatal education, or was this in the inclusions criteria that they must have attended education in the antenatal period? Also I notice you included primiparous and multiparous women in the study. Did more primiparous mothers attend classes than multiparous women? Did you look at the comparison between these groups. I would think there would be a difference in expectations and attendance between the two groups and this may alter the findings.

Author’s response: Unfortunately, we did not collect data on differences in attendance of antenatal classes in our study (and attendance of antenatal classes was not one of the inclusion criteria). We have added this fact to the list of methodological limitations in the discussion part of the revised manuscript (line 314).  We have performed additional analyses of differences in attitudes in satisfaction and attitudes toward antenatal educational based on parity, and we have found no statistically significant difference in these two factors between primiparas and multiparas.

Discussion

  • The title suggests there may be several factors during pregnancy and birth that are associated with PPD. The discussion seems to focus mostly on the need for antenatal education, which were statistically significant findings. The manuscript does not report on the informational and emotional support. I know these were not significant but they could still be compared to previous research. Often emotional support during birth is seen as positive and a lack of support a factor in postnatal depression. Were there differences in your study which may account for this conflict?

Author’s response: We agree with this reasonable comment. So, we added the respective paragraph into the Discussion: “The subjective perception of support during birth was perceived very positively in our group in all areas of support, i.e. informational, emotional and physical. It was rated best in the area of emotional support, but the differences were very small (table 2). Emotional support is one of the most important benefits provided by a support person for all types of assistance during birth. In particular, a feeling of security is important for a woman (Bohren et al., 2017), positively affects the experience of pain, ensures that women remain focused on childbirth (Lunda, P. et al. , 2018) and is assumed to have a significant role in maintaining mental well-being after birth (Iliadou, 2012). Most of the women from the group confirmed the presence of a support person who provided positive support during childbirth, but a significant association between birth support and PPD was not confirmed. However, it is possible to assume that social support during pregnancy, childbirth and in the postpartum period is generally a protective factor for development of PPD (Lunda et al., 2018; Nakmura et al., 2020; Tani, Castagna, 2016). However, support during childbirth is also involved in overall birth satisfaction, where the link with the risk of PPD has been confirmed (Urbanová et al., 2021).”(line 237)

  • Line 226. What is meant by ‘can provide pair….’ Please could you clarify.

Author’s response: We replaced this with “women as well as their partners”.

  • Line 233. Is there any research to back up the statement ‘too much time spent on postnatal education can result in negative effects’? Is this related to the findings of reference 27? If not is a bold statement to make without evidence.

Author’s response: The sentence. „too much time spent on postnatal education....“ is related to findings of the reference 27 ( McCarter-Spaulding , Shea, 2016), we placed this reference after the given sentence.

  • Line 235. I am unsure what you mean by ‘Pregnancy is psychological terminology in referred as a developmental crisis…’. Please could you clarify what you are saying.

Author’s response: Yes, this formulation would be misleading, we replaced the word “crisis” with “transition”.

  • Line 244. I think the work ‘eliminate’ is far too strong a word to use. To eliminate means remove completely. Antenatal education will not be able to eliminate PPD. Reduce would be a more appropriate word.

Author’s response: We agree, we replaced it with “reduced”.

Conclusion

  • As in the discussion, the conclusions does not refer to information and emotional support during labour.

Author’s response: We were dealing with this issue in our previous papers (Urbanova et al., 2021).

  • Line 291. ‘Therefore, midwives can soon….’. I feel this is quite a bold statement to make, can worsening depression be prevented by well-structured antenatal classes? You could mention that antenatal classes may be one way of reducing the symptoms or supporting women’s mental health.

Author’s response: We rephrased this sentence as: “Therefore, midwives can soon enough implement appropriate intervention as one of the ways of reducing the development of depressions potentially leading to further negative consequences” (line 333).

References

  • There are a large number of references that are over 5 years (the time period suggested by the journal). I understand the need for the use of seminal papers and references for screening tools which are used which may be decades old. Please could you review and update references where possible.

There are many references which could be updated, for example:

Adams, 2012. There many papers looking at mode of delivery and PPD. E.g. Baba eta l 2021     looked at mode of delivery and postpartum depression in Japan and a systematic review and meta analysis by Sun et al 2021 also reived this topic.

Hayes, 2001. There are many papers looking at antenatal education and perinatal depression. A recent article looked at antenatal education in primiparous women. Cankaya et al 2021.

Bergstrom, 2009. There are many papers with antenatal education on epidural rates. There has recently been a systematic review and meta-analysis. Hong et al 2021. Although this contradicts the paper you have referenced relating to epidural rates.

Author’s response: The list of references was updated.

Round 2

Reviewer 1 Report

The authors amended all of my comments. Thank you!

Author Response

Thank you for your helpful assistance!

Reviewer 2 Report

Thank you for the changes you have made.

I only have 2 comments and these are probably due to English translation and my be addressed by the type setters.

1) Line 130 says ..we addressed together 71..' do you mean 'we approached 710...

2) Lines 279-280 'it was rated best', I am not sure what this means

Author Response

1) Line 130 says ..we addressed together 71..' do you mean 'we approached 710...

We agree with this comment. We corrected it as suggested, i.e. "...we approached 710..."

2) Lines 279-280 'it was rated best', I am not sure what this means

We rephrased the sentence as: "...The area of emotional support was best rated...."